# Effect of Low-Frequency Renal Nerve Stimulation on Renal Glucose Release during Normoglycemia and a Hypoglycemic Clamp in Pigs

**DOI:** 10.3390/ijms25042041

**Published:** 2024-02-07

**Authors:** Marius Nistor, Martin Schmidt, Carsten Klingner, Caroline Klingner, Georg Matziolis, Sascha Shayganfar, René Schiffner

**Affiliations:** 1Orthopaedic Department, Jena University Hospital, 07747 Jena, Germanyg.matziolis@waldkliniken-eisenberg.de (G.M.); 2Institute for Biochemistry II, Jena University Hospital, 07747 Jena, Germany; martin.schmidt@med.uni-jena.de; 3Department of Neurology, Jena University Hospital, 07747 Jena, Germany; carsten.klingner@med.uni-jena.de (C.K.); caroline.klingner@med.uni-jena.de (C.K.); 4Emergency Department, Helios University Clinic Wuppertal, 42283 Wuppertal, Germany; sascha.shayganfar@helios-gesundheit.de; 5Faculty of Health/School of Medicine, Lehrstuhl für Klinische Akut- und Notfallmedizin, Witten/Herdecke University, Alfred-Herrhausen-Straße 50, 58448 Witten, Germany; 6Emergency Department, Otto-von-Guericke University, 39120 Magdeburg, Germany

**Keywords:** low-frequency renal nerve stimulation, hypoglycemia, glucose metabolism, diabetes mellitus

## Abstract

Previously, we demonstrated that renal denervation in pigs reduces renal glucose release during a hypoglycemic episode. In this study we set out to examine changes in side-dependent renal net glucose release (SGN) through unilateral low-frequency stimulation (LFS) of the renal plexus with a pulse generator (2–5 Hz) during normoglycemia (60 min) and insulin-induced hypoglycemia ≤3.5 mmol/L (75 min) in seven pigs. The jugular vein, carotid artery, renal artery and vein, and both ureters were catheterized for measurement purposes, blood pressure management, and drug and fluid infusions. Para-aminohippurate (PAH) and inulin infusions were used to determine side-dependent renal plasma flow (SRP) and glomerular filtration rate (GFR). In a linear mixed model, LFS caused no change in SRP but decreased sodium excretion (*p* < 0.0001), as well as decreasing GFR during hypoglycemia (*p* = 0.0176). In a linear mixed model, only hypoglycemic conditions exerted significant effects on SGN (*p* = 0.001), whereas LFS did not. In a Wilcoxon signed rank exact test, LFS significantly increased SGN (*p* = 0.03125) and decreased sodium excretion (*p* = 0.0017) and urinary flow rate (*p* = 0.0129) when only considering the first instance LFS followed a preceding period of non-stimulation during normoglycemia. To conclude, this study represents, to our knowledge, the first description of an induction of renal gluconeogenesis by LFS.

## 1. Introduction

The renal plexus forms a mesh-like network of nerves that surrounds and runs alongside the renal artery and comprises efferent and afferent sympathetic nerves from the minor splanchic and lumbal nerves [1,2,3]. Until quite recently, it was thought that the parasympathetic innervation of the kidneys, hailing from the truncus vagalis posterior, does not form part of the renal plexus. Recently though, van Amsterdam et al. were able to prove, based on a post-mortem histological investigation, the presence of parasympathetic nerves running alongside the renal artery as well [4]. Regarding the specific spatial patterns of sympathetic nerves around the renal artery in humans, there have recently been new insights that point toward a greater distribution within the anterior and superior quadrants, as well as a preponderance of sympathetic innervation around the right renal artery compared to the left [5].

The effects of renal sympathetic nerves are mediated through release of ATP, neuropeptide Y, vasoactive intestinal peptide, and, principally, norepinephrine [2,6]. The release of the aforementioned are, in turn, caused by the activation of adrenergic receptors (AR), namely alpha-1, alpha-2, beta-1, beta-2, and beta-3 adrenergic receptors; the distribution of which varies within anatomic structures, with alpha-1-AR primarily being found in the arterioles, alpha-2-AR in the proximal tubules, and beta-ARs along the nephron segments, as investigated, mainly, in a range of animal models [6,7,8,9,10]. Sensory nerves are primarily comprised of mechano-and chemoreceptors and modulate a number of central and local—through the reno-renal reflex—responses to varying states of intravascular pressures and changes in pH [2]. Sympathetic signaling regulates renal vascular tone, sodium, water, bicarbonate, and chloride reabsorption, as well as renal blood flow, the glomerular filtration rate (GFR), and renin secretion [2,6,11,12,13]. Evidence for the effects of renal sympathetic nerve activity (RSNA) has been accrued in a number of animal models, which for example, showed renin release and the modulation of sodium and water reuptake by low to mid-range frequency stimulation [14,15,16,17] and changes in renal blood flow (RBF) through vasoconstriction with high-frequency stimulation, though the exact mechanisms of RBF modulations and the influence of underlying oscillations in sympathetic tone on the renal vasculature remain unclear at this point [13,18,19,20,21].

Through maladaptive processes in response to chronic pathological states or defective signaling, chronic infections, or parenchymal damages, renal efferent sympathetic and sensory nerves are known to represent crucial contributing factors in the pathogenesis of many diseases, and are suspected to play a role in a further number whose pathophysiological foundation remain the subject of ongoing investigations. Prominently, these pathologies comprise, but are not limited to, hypertension, obesity, and insulin resistance, and consequently, diabetes mellitus, metabolic syndrome (as a consequence of the higher incidences of its singular pathologies), atrial fibrillation, cardiovascular disease, and renal inflammation and fibrosis itself [2,22,23,24,25]. Particularly due their share in the pathophysiology of arterial hypertension—amongst other mechanisms via a release of renin with subsequent increase in sodium and water reabsorption, decrease in renal blood flow and GFR, and therefore, ultimately a blood pressure elevation [13,26,27]—renal nerves have become a target for device-based interventions in treatment-resistant hypertension. Although in the now nearly two decades since its first application in clinical practice the results of studies on renal denervation in arterial hypertension have been ambiguous and, in some cases, controversial [28], the new generation of well-designed, large multicenter studies have been able to show a significant blood pressure reduction [29,30,31,32]. This led to a joint statement from the European Society for Cardiology (ESC) and the European Association of Percutaneous Cardiovascular Interventions that recommended (contrary to the ESC guidelines of 2018 [33]) renal denervation for certain patient groups; for example, those with treatment-resistant hypertension [34]. In subset analyses and animal studies, renal denervation has also been proposed as a potential treatment for other pathologies such as cardiovascular diseases and arrythmias [25,35,36,37,38], as well as insulin resistance, diabetes mellitus, and metabolic syndrome as a whole [39,40,41].

The observed beneficial effects of renal denervation on the glucose metabolism are not unexpected as sympathetic innervation of renal glucose release has been proposed for a number of years, though the exact mechanisms remain unknown. A number of decades ago, Greven et al. was able to show the stimulation of renal glucose release through exogenous epinephrine infusion [42]. More recently, Jiman et al. examined the effects of electrical stimulation on urinary glucose excretion in rats, reporting an increase following high-(kilohertz) frequency stimulation with a resulting blockade of action potentials, and a decrease in urinary glucose excretion without a change in urinary glucose concentration through low-frequency stimulation (LFS) [43].

The capacity of the kidneys—as the sole other organ beside the liver that can produce significant amounts of glucose—for gluconeogenesis is of particular importance in considering the potential benefits, but also risks, of a renal denervation procedure. In recent decades, recognition of the importance of renal gluconeogenesis for glucose homeostasis has increased and the concept of hepatorenal reciprocity—though not yet fully understood—has been established to describe the fact that the proportion of the liver and the kidneys in overall gluconeogenesis vary based on the current needs, energy demands, and metabolic state of the other organ(s), respectively [44,45]. Renal gluconeogenesis is modulated by receptor densities (SGLT1, SGLT2, and GLUT1), hormonal effects (cortisol, insulin, and catecholamines), and substrate availability (amino acids, lactate, etc.), but also by sympathetic nerves [46,47,48]. Our research group has previously shown that renal denervation halves the renal gluconeogenic response to a severe hypoglycemic episode in pigs [49]. Furthermore, we were able to show that a prenatal administration of dexamethasone in late-term pregnant sows induced both elevated baseline levels of ACTH and cortisol in the offspring and heightened the increase in these respective hormones during a hypoglycemic episode [50]. Interestingly, animals from the same litter who underwent a unilateral renal denervation procedure exhibited only a 25% decrease in the renal gluconeogenic response compared to the animals in the above-mentioned study that did not receive a prenatal glucocorticoid administration [51]. This might be indicative of early life adaptive processes that downregulate the share of sympathetic innervation in renal gluconeogenesis in favor of other regulatory mechanisms in order to maintain glucose homeostasis.

Based on our previous investigations, we set out to further investigate the effects of efferent sympathetic nerves on renal gluconeogenesis. Whereas our previous experimental model showed a reduction of renal gluconeogenesis after the ablation of renal nerves, we designed a positive-control study by innervating renal nerves with unilateral LFS and measuring side-dependent renal net glucose release (SGN) and comparing the results with the contralateral, non-stimulated organ.

## 2. Results

To investigate whether LFS alters SGN we subjected seven female pigs to LFS, emitted by a pulse stimulator, during normoglycemia and a subsequent mild hypoglycemic episode caused by a hyperinsulinemic–hypoglycemic clamp.

### 2.1. General Characteristics of the Experiment and Changes of Vital Parameters

Heart rate increased mildly from 104 ± 9 bpm at baseline to 113 ± 9 bpm during normoglycemia (which was preceded by surgical instrumentation) and moderately to 169 ± 22 bpm during the hypoglycemic period. Blood pressure increased slightly from 106/64 ± 5/5 mmHg to 114/73 ± 8/7 mmHg at the end of the hypoglycemic period. Body temperature decreased slightly from 37.8 ± 0.4 °C at baseline to 36.9 ± 0.3 °C at the end of the hypoglycemic period. Some information regarding the characteristics of the animals and basic vital parameters is provided in Table 1.

Hypoglycemia was established (as defined by our experimental conditions to be ≤3.5 mmol/L) after 90 min and was maintained for 75 min. Mean blood glucose decreased from 8.9 to 2.1 mmol/L (27 out of 35 individual test values were below 3.0 mmol/L) (Figure 1). Since there were slight variations in the time necessary to establish hypoglycemia, all subsequent measurements refer to the first time a blood glucose level below 3.5 mmol/L was measured. 

### 2.2. Effects of Examined Parameters in a Linear Mixed Model

We found that LFS significantly decreased urine volume (Figure 2, *p* = 0.0006). LFS showed a significant effect on the PAH concentration (*p* = 0.004), whereas hypoglycemia in itself and the interaction between LFS and hypoglycemia had no significant effect on PAH in a linear mixed model (Figure 3A). Median plasma concentrations of PAH decreased from 18 to 6 µg/mL (*p* = 0.0012) (Figure 3B). Neither hypoglycemia nor LFS had a significant effect on SRP (Figure 3C). LFS and hypoglycemia had a significant effect on GFR (*p* = 0.0176), while only hypoglycemia exerted significant effects on inulin clearance (*p* = 0.0033) (Figure 4). 

Notably, the main effect of ‘hypoglycemia’ was found to be statistically significant (t = 3.4292, *p* = 0.0010), indicating that the extent of ‘hypoglycemia’ is associated with a significant change in the SGN. However, the main effect of ‘electrical stimulation’ did not reach statistical significance (t = 1.6229, *p* = 0.1089) (Figure 5). The interaction term between electrical stimulation and hypoglycemia nonetheless showed a trend towards significance (t = −1.7898, *p* = 0.0776), implying a potential interaction effect between the two predictors. This suggests that the combined influence of electrical stimulation and hypoglycemia on renal net glucose release might differ from their individual effects, and this interaction effect warrants further investigation and interpretation.

LFS decreased sodium excretion significantly in the linear mixed model, both during normoglycemia and hypoglycemia (Figure 6). LFS caused no significant decrease in urinary flow rate in the same statistical model both during normoglycemic and hypoglycemic conditions (Figure 2). 

### 2.3. SGN, Sodium Excretion and Urinary Flow Rate in a Wilcoxon Signed Rank Exact Test

In a Wilcoxon signed rank exact test that only considered two time points in each animal, respectively, and the first occurrence of a period of LFS being followed by a preceding period of normoglycemia, and compared these with the contralateral, non-stimulated organ (for raw data see Table 2), LFS caused a significant increase in SGN (*p* = 0.03125) (Figure 7). Utilizing the same limitations and statistical model, LFS caused a significant decrease in sodium (*p* = 0.0017), and in urinary flow rate (*p* = 0.0129) as well (Figure 7). The Appendix A provides the raw data for side-dependent renal glucose release during the whole duration of the experiments.

## 3. Discussion

In this study we examined the effects of LFS during normoglycemia and mild hypoglycemia in pigs. In a linear mixed effects model that incorporated measurements and effects across all time points, electrical stimulation did not significantly change SGN. SRP remained unchanged in the linear mixed model during all experimental conditions, whereas LFS during hypoglycemia significantly decreased GFR. Consistent with prior investigations, LFS caused a decrease in urinary flow rate and sodium excretion in the linear mixed model [16,43]. Surprisingly, in the same statistical model, hypoglycemia significantly increased SGN (*p* = 0.001). In the linear mixed model, there was a general trend towards significance regarding the interaction effect of LFS and hypoglycemia as incorporated in an interaction term.

Nonetheless, in the Wilcoxon signed rank exact test, LFS did significantly increase SGN (*p* = 0.03) when comparing the first instance in which LFS followed a preceding period of normoglycemia without stimulation with the same time points in the contralateral, non-stimulated organ. Furthermore, sodium and urinary flow rate also decreased significantly in the Wilcoxon signed rank exact test that incorporated the same time points as described above.

As in our previous examinations, we saw a moderate drop in body temperature despite thermal management. Due to the mild nature of the hypothermia, we do not consider the drop in body temperature to have exerted meaningful effects, thus confounding the experimental results.

To our knowledge, this is the first study that has investigated total renal glucose output by measuring the difference of arterial and venous glucose concentrations as a marker of SGN and its response to LFS.

Based on the findings of our prior investigations in models of renal denervation and hypoglycemic clamp procedures in pigs, we expected an increase rather than a decrease in SGN due to hypoglycemia, and furthermore, an increase in SGN due to LFS. The latter could only be found in a Wilcoxon signed rank exact test that incorporated two distinct time points—the first period of LFS that followed one of non-stimulation during normoglycemia—but not in a linear mixed model. Formerly, we were able to show that unilateral renal denervation decreases SGN compared to the non-ablated kidney, while absolute renal glucose output still remained positive compared to normoglycemia [49], and that this decrease is halved in specimens with an altered hypothalamic–pituitary–adrenal axis (HPAA) due to late-gestational glucocorticoid injections [51]. Despite the very circumscribed results for the effects of LFS on SGN that reached significance, this study can in our estimation still be cautiously regarded as a positive control to our previous investigations of the influence of renal denervation on renal glucose release.

The choice of the pig animal model was partly based on the fact that our preliminary experiments on HPAA activation and renal denervation were carried out in the same animal model, and thereby ensured comparability [50,51], and furthermore, because the pig animal model is an established model used to carry out studies on metabolic processes, which, due to the similarities of the respective physiological processes, can be approximated to humans. In addition, it is also possible to imitate diabetic metabolic situations [50,51]. The position and shape of the kidneys in pigs also lend themselves to analogies with human anatomy [50,51].

Importantly, this current study utilized a somewhat different experimental setup than that of our previous investigations of the influence of sympathetic nerves on renal gluconeogenesis. In our previous studies, we defined hypoglycemia as either below 3 mmol/L [51] or below 2 mmol/L [49], preceded by, respectively, a 24-h food withdrawal prior to the commencement of the intervention. In this study, we decided on a closer approximation to more frequently occurring episodes of mild hypoglycemia. Rather than a 24-h food withdrawal, we conducted the experiments after a 4-h fasted period, which resembles clinical reality more closely. The main significant effect of “hypoglycemia” in the linear mixed model suggests that the extent of a reduction in blood glucose levels is associated with meaningful variations in the resulting renal net glucose release. The share of renal gluconeogenesis in overall glucose homeostasis is known to increase the longer a fast endures [46,48]. Additionally, even if compared to our previous investigations, we did not have to utilize glucose infusions to balance the extent of the induced hypoglycemia; the necessary, if lower, insulin dosages to induce a hypoglycemic clamp might have, in connection with the more attenuated hypoglycemic state, resulted in a different metabolic situation since insulin is known to suppress renal gluconeogenesis [52,53]. Correspondingly, the significant increase in SGN due to LFS in the Wilcoxon signed rank exact test in six of the seven animals was observed right at (or at least reasonably close to) the beginning of the experimental procedure. This corresponds with the period when the animals had only received medium dosages of insulin and were still closer to their respective baseline metabolic states. It is therefore likely that the effects of LFS we were able to observe correlate with time points when the anti-gluconeogenic effects of the insulin had not yet taken complete hold and the magnitude of the hypoglycemic stimulus was minor. Nonetheless, as it was our intention to create an experimental condition comparable to the clinical situation of insulin-induced hypoglycemia, at least in this study, we considered insulin injections to be inevitable. Furthermore, the advantage of a unilateral experimental model is that both the stimulated and non-stimulated kidney were subject to the same global physiological conditions, so that at least to an extent the confounding influence of insulin injections is mitigated by this fact.

It appears, therefore, that the mild hypoglycemic state employed in the experimental setup induced renal cells to take up glucose rather than caused gluconeogenesis, and that the induced hypoglycemia was not of sufficient severity in the pigs’ current metabolic state to induce renal gluconeogenesis. Nonetheless, the trend towards significance in the interaction term between LFS and SGN in the linear mixed model does indicate that the effect of electrical stimulation on renal net glucose release might depend on the presence of hypoglycemia, albeit of a severity that the experimental design did not replicate. Subsequent studies should, therefore, examine the effects of LFS in a variety of metabolic conditions; amongst them being more severe hypoglycemic states, and potentially, even in normoglycemia without any hypoglycemic challenge. This, though, barring a significant metabolic challenge from, for example, an extensive fast, might be difficult due to the circumscribed share of renal gluconeogenesis in the non-fasted individual [46].

It is, furthermore, conceivable that, due to the length and gradual decrease in the glucose levels, as well as the steady administration of insulin, at least in the late stages of the experimental procedure, a response of the HPAA stress system and its effector hormone cortisol, which is known to increase gluconeogenesis, could have served as a confounding influence [54,55]. Our experimental setup, however, was not designed to register system-wide adaptions to the hypoglycemic experimental conditions; for example components, of hepatorenal reciprocity (changes in hepatic gluconeogenesis or glycolysis) [45,56] or shifts in any of the other factors that are known to modulate renal gluconeogenesis such as substrate availability (lactate, amino acids, etc.), hormone levels (glucocorticoids and catecholamines), or receptor activity or density (GLUT1, SGLT1, and SGLT2) [46,48]. Furthermore, we did not measure urinary glucose excretion and urinary glucose concentration as carried out by Jiman et al. [43]. However, the decreases in urinary flow rate and sodium excretion we measured were in agreement with a number of previous studies that described similar responses to LFS [16,57,58,59,60], which suggests that the absolute reduction in urinary glucose (without a change in glucose concentration) reported by Jiman et al. is likely due to these antidiuretic effects [43].

Whereas high-frequency (kilohertz) stimulation of the renal nerves can be considered a (reversible and transient) equivalent to renal denervation [61,62,63], LFS has been shown to induce a multitude of physiological effects, such as activation of the Na^+^/H^+^ exchanger through activation of the AT1-receptor pathway [16], renin release [14,17], and changes in the renal vasculature [13,15,19]. Recorded responses of renal blood flow (RBF) to LFS have been more ambiguous across a multitude of studies; in a comprehensive review on the dynamic nature of renal sympathetic nerve activity (RSNA) and its influence on RBF control, Schiller et al. emphasizes the complexity of dynamically shifting RSNA pattern and underlying baseline oscillations of nerve activity [20]. We were only able to measure a significant increase in SGN in response to LFS if the stimulation was preceded by a period of non-stimulation during normoglycemia, and then only once. It is conceivable, therefore, that the one successful stimulation in each animal satisfied necessary underlying conditions in RSNA patterns, the specificities of which the experimental model was neither designed nor sufficient to uncover, or that subsequent stimulations disturbed physiological equilibrium. Due to the low frequencies utilized we did not anticipate a stabilization period between individual stimulations to be necessary, though retrospectively, this might have been the case and should be considered in subsequent investigations. Similarly, a depletion in transmitters cannot be ruled out as a reason for the subsequent failure of LFS to induce gluconeogenesis.

Further limitations of this study have to be noted in regard to the stimulation protocol. We used an automated randomization procedure, which entailed computer-automated assignation of the length and time point of LFS (performed autonomously by the pulse generator), which has the advantage of preventing investigator bias and allowed for a spread of “interventions” (LFS) over the periods of experimental conditions (normoglycemia and hypoglycemia). Nonetheless, this led to a certain disparity between individuals, which might have contributed to the wide inter-individual differences in the results. The exact distributional range of the LFS was similarly randomly and automatically computed by the pulse stimulator within the range of 2 Hz to 5 Hz. This protocol has the advantage of covering a range of frequencies of stimulation which therefore increases the likelihood of finding a frequency that most adequately replicates a physiological stimulus. Furthermore, it reduces the number of experimental animals required, since otherwise, to cover a range of frequencies, a much higher quantity of animals would be needed, if, as now appears likely, stabilization periods are necessary and an interference with, or disturbance of, the underlying transmission patterns of RSNA after a singular stimulation is to be ruled out. Nonetheless, the obvious disadvantage of this approach, especially considering the rather substantial periods of time (respectively 15 min) that resulted in a single value for the statistical analysis, lies in the impossibility of identifying a particularly efficacious frequency of stimulation.

Another conceivable confounding influence in a unilateral experimental model is an interference by the reno–renal reflex, first described in the 1980s [64]. Particularly in recent years, many new insights have been gained into the anatomical distribution of renal afferent nerves—running alongside the efferent sympathetic nerves in the renal plexus—as well as their contribution to a number of pathophysiological states such as hypertension [1,65,66]. Renal afferents play a crucial role in the reno–renal reflex, which, in rats, has been shown to cause increases in contralateral urine flow rate and urinary sodium excretion (in response to ipsilateral decrease in sodium excretion, thereby preserving homeostasis) as well as a decrease in efferent renal nerve activity through an increase in ipsilateral ureteral pressure and concomitant activation of mechanoreceptors; furthermore, Hermansson et al. reported a contralateral decrease in renal blood flow following an ipsilateral LFS of renal nerves [67]. The reno–renal reflex has been studied most intensively in rats [68], but has also been demonstrated in cats [69,70] and dogs [71]. To our knowledge, there has only been one study performed in pigs, which concluded that pelvic distension and the hyperperfusion of one kidney showed no evidence for a reno–renal reflex, considering diuresis of the contralateral organ as an outcome parameter [72]. A confounding of the results through a reno–renal interaction can nonetheless not be wholly ruled out, especially since our previous results were based on an experimental setup utilizing renal denervation, which, since we did not deliberately spare or identify afferent nerval tissues, would have presumably not faced this particular variable. Whether there is a reno-renal interaction regarding sympathetic activation of renal gluconeogenesis, and if so, to what extent and under what conditions, remains nonetheless, to our knowledge, unknown.

One of the main pitfalls, or weaknesses, of the mathematical model we employed to compute SGN is the necessity of incorporating SRP in the equation alongside the measured arterio–venous glucose difference. Since urine volume over time is one of the variables necessary to compute SRP, we have found in this study, as in our previous investigations utilizing the same equations [50,51], a strong confounding influence of the inter-individual differences in urinary flow rate due to individual physiologies. This resulted in large standard deviations and a distinct vulnerability of the computed SGN to said variable. Nonetheless, hyperperfusion was necessary to avoid the administration of catecholamines, which would have obfuscated the examined physiological mechanisms to an even greater extent. One possible way to avoid such vulnerability in the examined parameter, SGN, is a different method of computation or detection. Gluconeogenesis can be measured by the administration of a variety of tracers and isotopes; though, until now, there is no gold standard, and these methods still allow only an approximation to physiological mechanisms and exact values due to the complexity of potential confounding variables (glycolysis, etc.), and often require complex imagery such as nuclear magnet resonance spectroscopy [73].

Beside these ineluctable variables, we strove to isolate further confounding factors by standardizing, for example, the size and age of the pigs and the experimental conditions (such as heat maintenance and infusion protocols), and to prevent investigator bias by utilizing the formerly described fully automated stimulation protocol.

In our opinion, unilateral denervation does not need to be stratified for a side-separated consideration of renal glucose output because the statistical analysis also takes the side difference into account. Nonetheless, in addition to side-separated stimulation, alternating simultaneous or non-simultaneous stimulations could be investigated in further studies to conclusively rule out unilateral differences as a potential confounding factor.

## 4. Methods

### 4.1. Experimental Animals, Surgical Procedures

The Saxony animal welfare committee (Leipzig; permission number TVV 64/15 from 12 February 2015 to 5 May 2021) approved the experimental procedures as described below. The animals in this study were all pigs of the German landrace and were reared in a conventional agricultural holding facility. The animals ranged in weight from 32 kg to 41.5 kg. All experimental and surgical procedures were conducted in strict accordance to local standards and the “Guide for the Care and Use of Laboratory Animals” [74].

Immediately before the experiments commenced the animals underwent a four-hour food withdrawal, albeit with ad libitum access to water. The surgical procedure was performed under general anesthesia, in supine position and under strict sterile conditions. Anesthesia was induced by intramuscular injection of 15 mg/kg ketamine hydrochloride (Ketavet^®^, 100 mg/mL, Pharmacia Upjohn, Erlangen, Germany) and 0.2 mg/kg midazolam hydrochloride (Midazolam-ratiopharm^®^, Ulm, Germany). Thereafter, 0.2–0.3 mg/kg propofol (Disoprivan^®^, AstraZeneca, Wedel, Germany) was administered through a venous catheter in the ear vein (Vasocan^®^, Braun Melsungen, Melsungen, Germany), followed by orotracheal intubation (Trachealtubus, Rüsch, Kernen, Germany). Subsequent anesthetic maintenance was achieved through continuous inhalation of 1.5% isoflurane (Isofluran^®^, DeltaSelect, Dreieich, Germany) and O_2_; analgesia was maintained via the application of 0.003 mg/kg fentanyl per hour (0.05 mg/mL Fentanyl, Janssen). Dosages of up to 0.1 mg/kg pancuronium (Pancuronium-Actavis^®^, Actavis, München, Germany) were administered for muscular relaxation. Corneal moisture was ensured with the application of eye drops (Corneregel^®^, Bausch&Lomb, Berlin, Germany). To ensure a stable blood pressure during anesthesia, the animals received isotonic saline infusions (Isotonische Kochsalzlösung^®^, Fresenius, Bad Homburg, Germany). The carotid artery (Arteriofix^®^, Braun, Melsungen, Germany) and the jugular vein (Certofix Trio^®^, Braun, Melsungen, Germany) were catheterized for blood sampling and blood pressure measurement, as well as for intrapoperative administration of fluids and drugs. A midline laparotomy was performed to gain abdominal access, followed by an opening of the retroperitoneum. For urine sampling, size 6 charier catheters (Actreen^®^ Glys Cath, Braun, Melsungen, Germany) were inserted into both ureters. Local blood sampling was facilitated via bilateral instrumentation of the renal veins with vascular catheters (Certofix^®^ Trio, Braun, Melsungen, Germany), which were kept in position with Liquiband^®^ Flow Control (Advanced Medical Solution, Devon, UK). To maintain body temperature, we used a water mat system (Hico-Variotherm 555^®^, REF 550025; PFM Medical Hico gmbh, Köln, Germany); additionally, the animals were covered with sheets where feasible. A simplified graphic representation of the surgical instrumentation is provided in Figure 8.

### 4.2. LFS

A nerve cuff electrode (supplied with isolated pulse stimulator, see below) was placed around the plexus renalis without constricting or exerting pressure on the renal artery. The electrode was connected to an isolated pulse stimulator (Model 2100, A-M Systems, Loop Sequim, WA, USA). The stimulation amplitude was fixed at 10 V. For low frequency stimulations, the isolated pulse stimulator was set to vary automatically between 2 Hz and 5 Hz, generating biphasic pulses. The stimulation pulse and amplitude were fixed at 0.5 msec and 10 V. The side of the stimulated kidney had been determined via a randomization list. Stimulation periods were similarly randomly ordered between trials across all experiments in order to mitigate sequential effects. The duration of the individual stimulation periods was 15 min. Further episodes were randomly determined by coin toss. Due to the low frequencies utilized, we deemed no stabilization periods necessary between individual values.

### 4.3. Induced Polyuria and Maintenance of Blood Pressure

To avoid obfuscation of the examined physiological mechanism and to avoid the necessity of administration of exogenous catecholamines it was expedient to submit the kidneys to a hyperperfusion, which was achieved with high volume infusions of sterile isotonic saline with a subsequent polyuria exceeding 100 mL/h/kidney. The infusion rates were adjusted according to the invasively measured blood pressure as well as urine production, and varied between 1 and 2 L/h. With this method, no animals exhibited a significantly decreased urine volume, and the administration of furosemide was not necessary in any of the experimental animals.

### 4.4. Hypoglycemic Clamp

Hypoglycemia was established via administration of a bolus of 10 IU human regular insulin (Actrapid^®^, Penfill^®^, 100 IU/mL, Novo Nordisk Pharma, Mainz, Germany). Hypoglycemia was defined as a blood glucose level below 3.5 mmol/L. In a mildly fasted state after four hours of food withdrawal, we aimed for a slow drop in the blood glucose levels. At intervals of 7.5 min, arterial blood glucose levels were monitored with a blood glucose meter (Contour^®^, Bayer AG, Leverkusen, Germany). If necessary, further boli of insulin were administered to induce hypoglycemia as formerly defined; individual insulin dosages expedient to achieve the desired range varied between 10–60 IU of insulin (per experimental procedure, respectively).

### 4.5. Determination of Side-Dependent Glomerular Filtration Rate, Renal Plasma Flow and Gluconeogenesis

Through a two-step process—which consisted of an initial bolus of 3.1 g para-aminohippurate (PAH) (Sigma-Aldrich, Taufkirchen, Germany) and 0.55 g inulin (Sigma-Aldrich, Taufkirchen, Germany), followed by a continuous infusion consisting of 3.3 g PAH and 0.75 g inulin in a solution of sterile isotonic saline at a rate of 250 mL/h until the end of the experimental procedure—PAH and inulin were adjusted to be near equilibrium. An in-depth explanation of the procedure utilized is provided in our previous study [49].

Side-dependent renal plasma flow (SRP) was determined by use of the equation: SRP [mL/min] = C_UrinePAH_ [mg/L] × V_UrineVolumeOverTime_ [mL/min]/C_PlasmaPAH_ [mg/L]/ 0.9 [75,76], for each kidney, respectively. For the analysis of C_UrinePAH_, urine specimens were collected from each ureter via the established catheters. Blood specimens were drawn from the carotid artery for analysis of C_PlasmaPAH_. Specimens of blood and urine analysis were collected, respectively, every fifteen minutes. Urine collected from the ureter catheters was used to measure V_UrineVolumeOverTime_; urinary volume was collected and analyzed every fifteen minutes. GFR was calculated for each kidney via the following equation: C_UrineInulin_ [mg/L] × V_UrineVolumeOverTime_ [mL/min]/C_PlasmaInulin_ [mg/L] [77]. Urine specimens for the analysis of C_UrineInulin_ were collected from the catheters in both ureters. Blood specimens drawn from the carotid arteries were used for determination of C_PlasmaInulin_. SGN was determined by use of the following equation: SGN [mmol/min] = SRP [L/min] × (C_VenousGlucose_ [mmol/L] − C_ArterialGlucose_ [mmol/L]). To measure C_VenousGlucose_, blood specimens from the catheterized renal arteries were used. For the analysis of C_ArterialGlucose_, blood specimens were drawn from the carotid artery.

### 4.6. Pre-Analytical Methods

After collection, all plasma urine samples were aliquoted and stored until analysis. Samples were thawed in a water bath and subsequently centrifuged and vortexed in a microcentrifuge at room temperature to determine PAH. The resulting cleared supernatants were utilized for additional analyses. All reagents were of analytical grade (acquired from Roth, Karlsruhe, Germany).

### 4.7. Quantitation of Inulin

To allow for the use of a microplate reader, Roe et al.’s [78] quantitation method was adapted by us according to the following protocol: a ten minute sample centrifugation was succeeded by preparation of routinely diluted (1:10; urine) and undiluted (plasma) samples for each animal and site of specific sample, respectively. A comprehensive explanation of this quantitation method is supplied in our previous study [49].

### 4.8. Quantitation of PAH

For quantitation of PAH, we utilized the established method as described by Agarwal et al. [79]; briefly, this refers to the use of a microplate assay relying on the reaction of *p*-dimethylaminocinnamaldehyde (Sigma-Aldrich, Taufkirchen, Germany) with PAH, which provides results that conform to a large degree with HPLC-based methods. A comprehensive description of the method mentioned above is supplied in our previous study [49].

### 4.9. Quantitation of Sodium Excretion

To determine sodium concentrations from urine samples, we utilized a routine clinical procedure.

### 4.10. Statistical Analyses

For descriptive statistics summarizing the outcome parameters of the different measurements, means ± standard deviation [SD] were used if all data sets were normally distributed. Where at least one data set did not follow a normal distribution, box plots were used. A linear mixed model was adapted to investigate the influence of the parameter “hypoglycemia” and “electrical stimulation” on the renal gluconeogenesis of the stimulated kidney. To detect possible interactions between the electrical stimulation and hypoglycemia, an interaction term for both parameters was included in the model as well. Given the presence of repeated measurements from the same animals, we incorporated a random effects structure to account for potential inter-animal variability. Specifically, a random intercept was specified for each combination of animal and measurement number. All analyses were performed with R 4.0.5. The linear mixed- models were fitted using the lme() function from the nlme package in R. The fitted models allowed us to estimate the coefficients associated with the main effects and interaction terms, along with their corresponding standard errors and *p*-values. The Appendix A provides comprehensive data of the linear mixed models. Additionally, a Wilcoxon signed-rank test was performed to assess the differences between paired samples. The significance level was set to a = 0.05.

## 5. Conclusions

To our knowledge, this is the first study to investigate the effects of LFS on SGN. In a linear mixed model, LFS did not significantly alter SGN either during normoglycemia or hypoglycemia, though there was a general trend towards the statistical significance of combined LFS and hypoglycemia. Hypoglycemia significantly lowered SGN, which might be due to the general experimental conditions which only consisted of a mild hypoglycemic episode, and due to interferences of the necessary injection of insulin for the hypoglycemic clamp procedure, which, conceivably, could also be the reason for the absence of an effect of LFS on SGN. Further studies should analyze LFS in different experimental conditions that do not entail the same confounding factors encountered by us; for example, during different metabolic conditions, hypoglycemic conditions of greater severity, or utilizing different means of calculating SGN, such as using tracers or isotopes, which are not as vulnerable to inter-individual differences in basic physiologic parameters.

Nonetheless, we were able to demonstrate a significant increase in SGN due to LFS in a Wilcoxon signed rank exact test that only considered the first instance of LFS followed on a preceding period of non-stimulation during normoglycemia, which might be cautiously considered as a positive control of our prior investigation on the influence of sympathetic nerves on renal gluconeogenesis utilizing renal denervation. Nonetheless, further studies are necessary to confirm and expand on these results and to elucidate the underlying physiologic mechanisms.

## Figures and Tables

**Figure 1 ijms-25-02041-f001:**
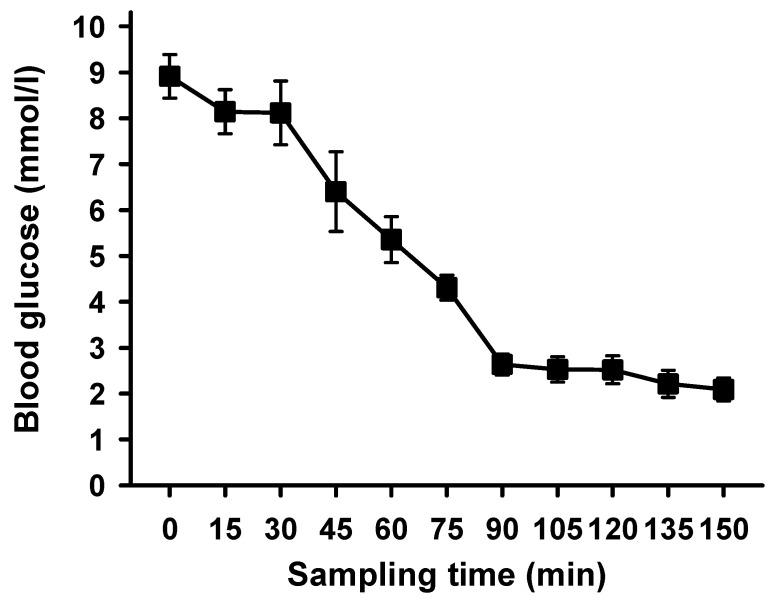
**Mean blood glucose during hypoglycemic clamp after unilateral stimulation of renal nerves.** Mean blood glucose [mmol/L] measurements were performed every 15 min; means ± SD, *n* = 7.

**Figure 2 ijms-25-02041-f002:**
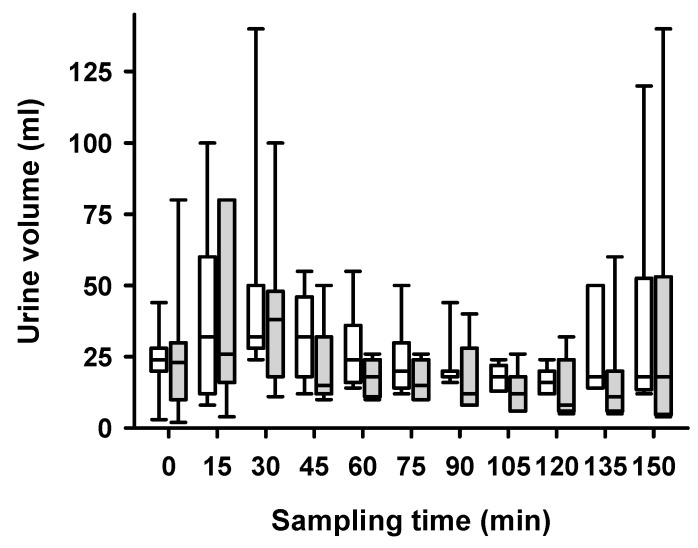
**Side-dependent urine volume over time after unilateral stimulation of renal nerves.** The side-dependent urine volume per 15 min interval during a hypoglycemic clamp with intermittent unilateral low-frequency stimulation (LFS) of the renal nerves. White boxes represent the non-stimulated side while grey boxes represent the stimulated side. Side-dependent urine production was measured during a high-volume infusion inducing hyperperfusion of the kidneys. Samples for measurements were taken every 15 min. Data are presented as box plots, showing medians, 25/75th percentiles (boxes), and 10/90th percentiles (whiskers), *n* = 7. The linear mixed model showed a significant effect of LFS (*p* = 0.0006) and hypoglycemia (*p* = 0.03), while no significant interaction (*p* = 0.80) was found.

**Figure 3 ijms-25-02041-f003:**
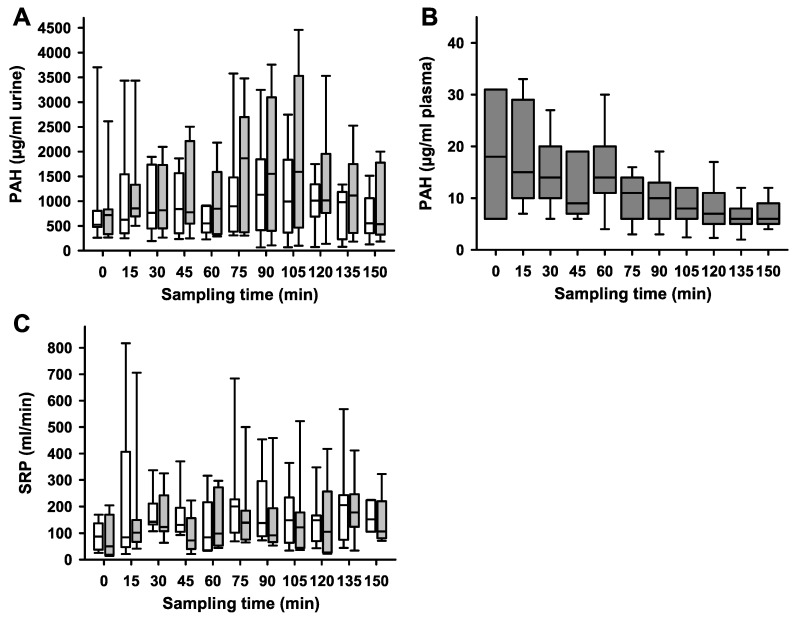
**Para-aminohippurate (PAH) (A) in urine; (B) in plasma and side-dependent renal plasma flow (SRP) (C) responses to unilateral renal nerve stimulation:** The figure presents data from experiments measuring PAH and SRP during hyperperfusion of the kidneys induced by a high-volume infusion protocol. Samples were collected every 15 min under two conditions: normoglycemia (0 to 75 min) and hypoglycemia (blood glucose ≤ 3.5 mmol/L, from 90 to 150 min). White boxes represent the non-stimulated kidney; grey boxes the stimulated side. Panel A represents PAH concentrations. Statistical analyses showed that LFS significantly increased PAH, while neither hypoglycemia nor the interaction of hypoglycemia and LFS had a significant effect. Panel B illustrates that plasma PAH concentrations obtained from carotid artery samples remained below the saturation threshold for tubular secretion during the whole experimental procedure. Panel C represents the SRP [mL/min] throughout the experiment. Neither LFS, hypoglycemia, nor their interaction term significantly altered SRP [mL/min]. Data are presented as box plots, illustrating medians, 25th/75th percentiles (boxes), and 10th/90th percentiles (whiskers). The sample size for these experiments was *n* = 7, with a significance level of *p* < 0.05.

**Figure 4 ijms-25-02041-f004:**
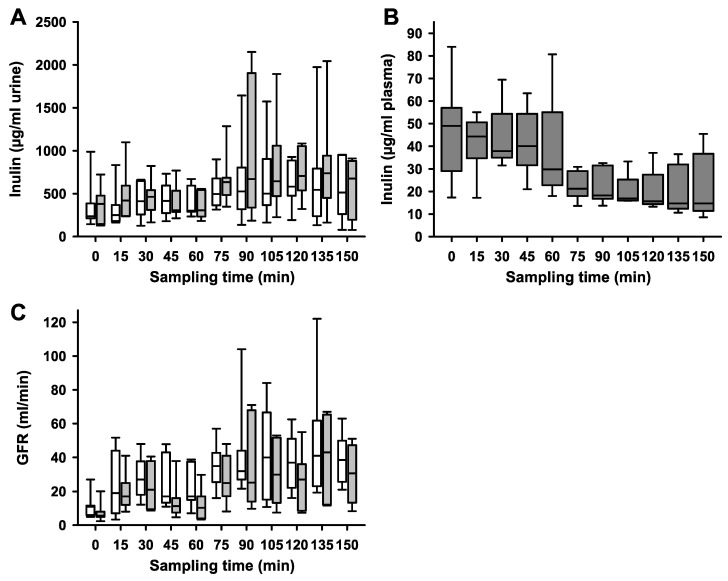
**Inulin (A) in urine; (B) in plasma and glomerular filtration rate (GFR) (C) after unilateral stimulation of renal nerves.** Samples for GFR measurements were taken every 15 min during high-volume-infusion-induced hyperperfusion of the kidneys during normoglycemia (0 to 75 min sampling time) and during hypoglycemia (blood glucose below 3.5 mmol/L, 90 to 150 min sampling time) was established. (**A**) Inulin [μg/mL] in urine during experiments duration in the non-stimulated (white boxes) and the stimulated (grey boxes) side (*p* = 0.23). (**B**) Plasma inulin concentrations in samples drawn from the carotid artery during the experiment are well below the saturation concentration for the tubular secretion system. (**C**) Decrease in GFR [mL/min] of the stimulated (grey boxes) side compared to the non-stimulated (white boxes) side (*p* = 0.0176) during LFS (not depicted). Data are presented as box plots, showing medians, 25/75th percentiles (boxes), and 10/90th percentiles (whiskers), *n* = 7.

**Figure 5 ijms-25-02041-f005:**
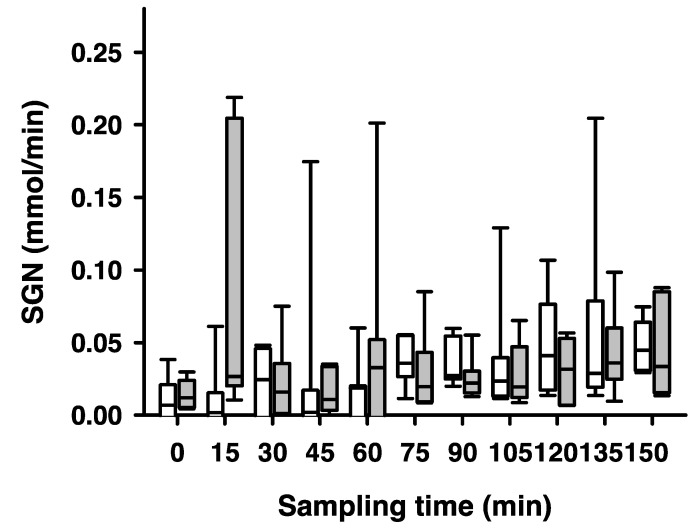
**Side-dependent renal net glucose release (SGN) after unilateral stimulation of renal nerves.** Samples for SGN measurements were taken every 15 min during high-volume-infusion-induced hyperperfusion of the kidneys during normoglycemia (0 to 75 min sampling time) and during hypoglycemia (blood glucose below 3.5 mmol/L, 90 to 150 min sampling time). White boxes represent SGN [mmol/min] of the non-stimulated kidney; grey boxes the stimulated kidney. No significant effect of SGN was observed in a linear mixed model (*p* = 0.11). Data are presented as box plots, showing medians, 25/75th percentiles (boxes), and 10/90th percentiles (whiskers), *n* = 7.

**Figure 6 ijms-25-02041-f006:**
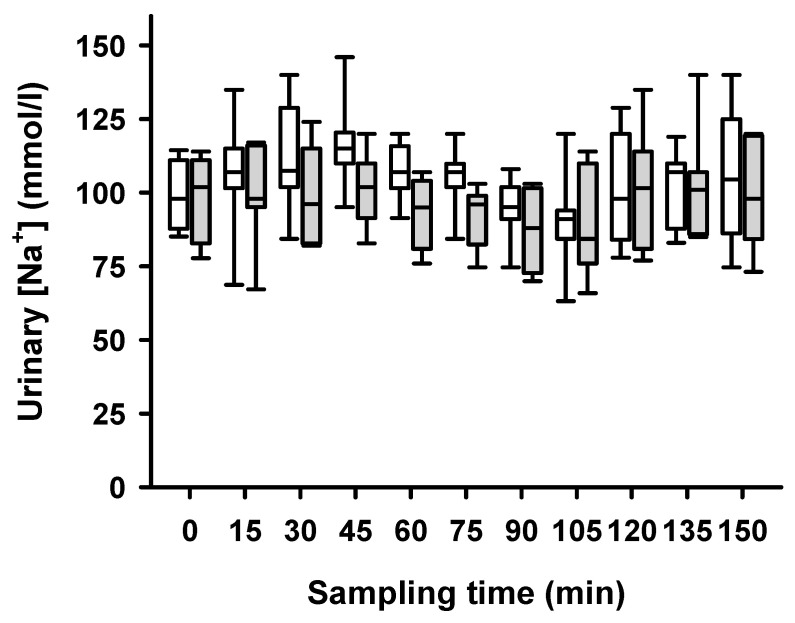
**Urinary sodium concentrations after unilateral stimulation of renal nerves.** Samples for sodium excretion measurements were taken every 15 min during high-volume-infusion-induced hyperperfusion of the kidneys in normoglycemia (0 to 75 min sampling time) and hypoglycemia (blood glucose below 3.5 mmol/L, 90 to 150 min sampling time). Decrease in urinary sodium [mmol/L] during the experiments in the non-stimulated (white boxes) and the stimulated (grey boxes) side (*p* < 0.0001). Data are presented as box plots, showing medians, 25/75th percentiles (boxes), and 10/90th percentiles (whiskers), *n* = 7.

**Figure 7 ijms-25-02041-f007:**
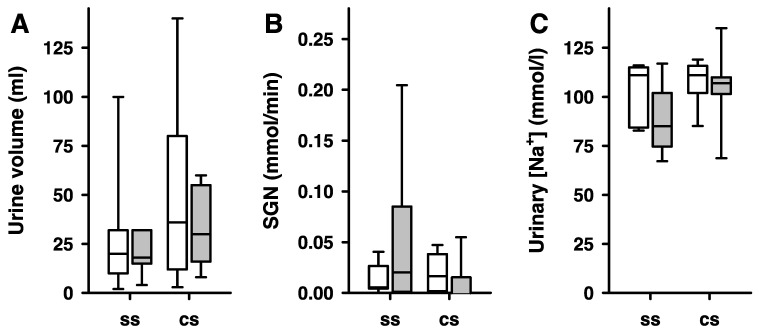
**Side-dependent urine volume, SGN and urinary sodium concentration in a single unilateral stimulation of renal nerves.** (**A**) Significant decrease in urine volume during normoglycemic LFS (first instance that LFS succeeded non-stimulation during normoglycemia) between the stimulated side (ss) and the contralateral side (cs) (Wilcoxon test; *p* = 0.0129). (**B**) The same experimental conditions as described above significantly increased SGN of the stimulated side (ss) compared to the contralateral side (cs) (Wilcoxon test; *p* = 0.03125). (**C**) The same experimental conditions as described above significantly increased urinary sodium of the stimulated side (ss) compared to the contralateral side (cs) (Wilcoxon test; *p* = 0.0017). Samples for measurements were taken over 15 min. Data are presented as box plots, showing medians, 25/75th percentiles (boxes), and 10/90th percentiles (whiskers), *n* = 7.

**Figure 8 ijms-25-02041-f008:**
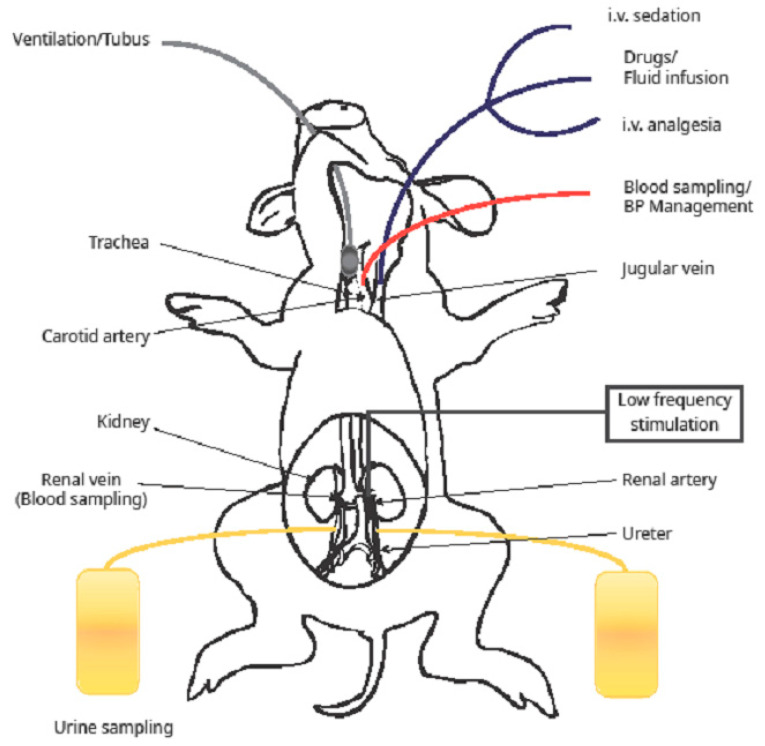
**Illustration of experimental setup.** The jugular vein was cannulated for saline and glucose infusion. A nerve cuff electrode was placed along the renal nerves of the right or left kidney and connected to a stimulation generator. The ureters were cannulated bilaterally in order to collect the urine separately.

**Table 1 ijms-25-02041-t001:** Comprehensive data on the vital parameters of the animals during the experimental procedure.

Variables	Parameters
Sex (n_male_/n_female_)	0/7
Age (days)	85 ± 7
Weight (kg)	37 ± 4
Weight of left kidney (g)	97 ± 18
Weight of right kidney (g)	97 ± 21
Heart rate at baseline (bpm)	104 ± 9
Heart rate during normoglycemia (bpm)	113 ± 9
Heart rate at the end of hypoglycemia (bpm)	169 ± 22
Blood pressure at baseline (systolic/diastolic) (mmHg)	106/64 ± 5/5
Blood pressure at the end of hypoglycemia (systolic/diastolic) (mmHg)	114/73 ± 8/7
Body temperature at baseline (°C)	37.8 ± 0.4
Body temperature at the end of hypoglycemia (°C)	36.9 ± 0.3
Data are given as means ± SD, *n* = 7	

**Table 2 ijms-25-02041-t002:** Original data for urine volume, SGN, and urinary sodium after a single unilateral stimulation of renal nerves.

Number of Animal							
	1	2	3	4	5	6	7
Body site of stimulation							
	left	left	left	left	left	left	left
Urine volume of stimulated kidney (mL)							
Before stimulation	10	2	10	20	28	32	100
After 15′ of stimulation	16	4	15	18	26	32	32
Urine volume of non-stimulated kidney (mL)							
No-stimulation	44	3	36	12	20	80	140
No-stimulation after 15′	8	16	30	28	60	40	55
SGN of stimulated kidney (mmol/min)							
Before stimulation	0.00536	0.0239	0.0049	0.0266	0.0043	0.0405	−0.0974
After 15′ of stimulation	0.02024	0.2046	0.0850	0.0012	0.0105	0.0356	0.0
SGN of non-stimulated kidney (mmol/min)							
No-stimulation	0.03824	−0.0686	0.0194	0.0166	0.0068	0.0017	0.0472
No-stimulation after 15′	−0.00084	−0.0692	0.0548	−0.0343	0.0169	−0.0664	0.0021
Urinary sodium of stimulated kidney (mmol/L)							
Before stimulation	114	82.8	84,4	116	111	98	115
After 15′ of stimulation	97	67.2	74.7	85	117	82	102
Urinary sodium of non-stimulated kidney (mmol/L)							
No-stimulation	114.4	85.1	115.8	107	111	102	119
No-stimulation after 15′	101.5	68.7	107.1	107	135	102	110

## Data Availability

The original contributions presented in the study are included in the article/Appendix A, further inquiries can be directed to the corresponding author.

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
