# Peer review of "Effect of Low-Frequency Renal Nerve Stimulation on Renal Glucose Release during Normoglycemia and a Hypoglycemic Clamp in Pigs"

_ijms, 2024, doi:10.3390/ijms25042041_

Round 1

Reviewer 1 Report

Comments and Suggestions for Authors

In this article, the authors followed their previous work and examined the unilateral low-frequency stimulation of the renal plexus during hypoglycemia and normoglycemia. The article is well-written, and the topic is interesting. Minor revision is suggested. 

1.        Can the authors provide a more comprehensive rationale for choosing pigs as the experimental model?

2.        Can the authors provide more  details of the pulse generator (2 to 5 kHz) used for renal denervation? How was the frequency range determined, and what considerations were taken into account?

3.        In this work, how did the authors address potential sources of error, and what steps were taken to ensure the precision and accuracy of these measurements?

4.        How applicable are the findings of this study to human physiology and pathophysiology, considering the inherent differences between pig and human renal systems? 

Author Response

Reviewer 1:

In this article, the authors followed their previous work and examined the unilateral low-frequency stimulation of the renal plexus during hypoglycemia and normoglycemia. The article is well-written, and the topic is interesting. Minor revision is suggested.

Answer: We would like to thank the reviewer for the evaluation of the manuscript and for its kind reception.

  1. Can the authors provide a more comprehensive rationale for choosing pigs as the experimental model?

Answer: Thank you very much, the comments and questions are appropriate. We had not yet addressed our reasons for the selection of the animal model.

. The background was that the animal model pig is an established animal model for carrying out metabolic processes with similarities to humans. It is also possible that diabetic metabolic situations can be imitated. At this point it should be noted that the animal model pig is also suitable for investigating prenatal influences in relation to fetal hpa-axis changes. The position, shape and metabolic processes of the kidney in pigs are also suitable for studies based on analogies with humans.

Reaction: We have added information pertaining to our reasoning to the discussion in lines 406-412.

  1. Can the authors provide more details of the pulse generator (2 to 5 kHz) used for renal denervation? How was the frequency range determined, and what considerations were taken into account?

Answer: The generator used is capable of providing pulsed stimulation in a certain predefined frequency range. This does not achieve renal denervation. However, high-frequency stimulation in the kilohertz range can be used to simulate the effects of renal denervation. As we carried this out in preliminary experiments, we deliberately did not choose high-frequency stimulation. Since low-frequency stimulation in the frequency range of 2 to 5 kHz can be deemed comparable to the dynamic nature of renal sympathetic nerve activity and there was a preliminary study (Jiman et al) in which glucose urinary excretion was studied under renal stimulation, we chose this particular frequency range. The generator imitates a large frequency range, which makes it possible to narrow down the necessary frequencies in an experiment to obtain physiological effects. We believe that these aspects have been appropriately addressed in the discussion. At this point we’d like to refer especially to lines 487-505.

  1. In this work, how did the authors address potential sources of error, and what steps were taken to ensure the precision and accuracy of these measurements?

Answer: This is a very important question, especially considering the considerable variation in our individual values. We already explained in the article why, for example, we did not utilise certain compounds or techniques – such as the administration of catecholamines or diuretics – that would directly influence some of the examined parameters. Furthermore, we already expanded on our reasoning for the non-fasted state of the animals (which poses a closer approximation to a physiological state than the experimental conditions of our prior studies of renal denervation).

Response: We nonetheless added a further couple of sentences underlining our efforts of standardising the experimental model and mitigating potential confounding factors in lines 583-594.

  1. How applicable are the findings of this study to human physiology and pathophysiology, considering the inherent differences between pig and human renal systems?

Answer: That is a good and crucial question. We do not know this in its entirety and can only speculate. As we point out in the article, the experimental setup is suitable for creating a situation in which these physiological effects can be demonstrated for the first time. Furthermore, we have already critically discussed the pitfalls of the experimental models and certain confounding influences. Ultimately, it can be assumed that the effects shown also exist in human physiology. However, we cannot prove this. This should be addressed in future studies.

We addressed the issue of (clinical) transferability in more depth in our prior investigation of the effects of renal denervation on the renal capacity for gluconeogenesis during hypoglycemic states  - as there might be certain inherent risks for patient groups undergoing this procedure who are at simultaneous risks for hypoglycemia (from antidiabetic medications). This current study we intended to serve as a positive control study of our prior investigations and to have, firstly, a more elementary research focus.

Reviewer 2 Report

Comments and Suggestions for Authors

This report describes an interesting experiment that aims to ask a simple and apparently novel question,  ie "What is the effect of low frequency renal nerve stimulation on renal glucose generation, at baseline and during hypoglycaemic clamping, over 150 minutes?"

I think that the paper could be presented in a clearer manner for those readers who are familiar with the basic concepts but do not have specialised knowledge.

he background information is extensive, and it is not clear that all the comparisons with previous work are relevant to the discrete experiment.

I would like to see explanations of the following -

1. What is the relevance of the observations on the offspring of sows who received pre-delivery dexamethasone to the current experiment?

2. How different are the glucose outputs from the L and R kidneys at baseline? Does the unilateral denervation need to be stratified for this, with the experiment repeated on the contralateral side?

3. Can the effects of insulin on the kidney be reliably dissociated from the effects of hypoglycaemia?

4. Given that the standard errors are quite high, how comparable are results with results of former experiments?

5. Statistical methods could be better described, with p values & tests supplied for parameters that are stated to increase or decrease

6. Comment re 150 min timeframe in the context of what is happening in the kidney - eg if steroid hormone alterations are relevant, this is a v short time.

7. Comment on potential clinical application of findings, if relevant

Comments on the Quality of English Language

English is good, but many sentences are very long and difficult to follow - I think the language could be clarified with overall improvement 

Author Response

Reviewer 2:

This report describes an interesting experiment that aims to ask a simple and apparently novel question,  ie "What is the effect of low frequency renal nerve stimulation on renal glucose generation, at baseline and during hypoglycaemic clamping, over 150 minutes?"

I think that the paper could be presented in a clearer manner for those readers who are familiar with the basic concepts but do not have specialised knowledge.

he background information is extensive, and it is not clear that all the comparisons with previous work are relevant to the discrete experiment.

Response:

We would like to thank the reviewer for their careful reading of the manuscript and constructive criticism.

As for the latter two points pertaining to the ( supposedly too expansive) length of the introduction/manuscript and the simultaneous wish for a more in-depth explanation of the rather specialized content – we would argue that there is a certain incoherence in this demands, or rather difficulty for us to cater to both of the reviewer’s wishes at once.

As the reviewer correctly noted the topic is rather specialized and we therefore tried to give ample and sufficient introductory scope for the reader that might not be very well acquainted with the topic, while also keeping in mind the expediency for clarity and brevity. We rather tend to think that we’ve accomplished this balance.

Since the second reviewer’s seems to agree with our own assessment, we’d rather refer the decision to the editor than significantly change the scope or content of our introduction.

Pertaining to the reviewer's point about the article containing many long sentences - we have broken up some of the longer, more complicated sentence structures throughout the manuscript and furthermore corrected a number of smaller mistakes/typos that, unfortunately, we didn't notice before. 

I would like to see explanations of the following -

  1. What is the relevance of the observations on the offspring of sows who received pre-delivery dexamethasone to the current experiment?

Answer: As described in the introduction, we saw a reduction of 50% in non-HPA-altered specimens and a 25 reduction of renal blood glucose mobilization after changes in the HPAA axis in pigs after renal denervation. We were able to show in 2 experimental setups that there is a dependence and variability of renal blood glucose regulation on the HPA axis. Our present experiment has developed out of these findings and we believe that these references to our prior work are therefore relevant both as an explanation of the context of this current study as well as to elucidate the later discussion of the deviations (and its consequences) from our prior experimental protocols in the discussion. For this, we’d like to refer the reviewer to lines 427-443.

  1. How different are the glucose outputs from the L and R kidneys at baseline? Does the unilateral denervation need to be stratified for this, with the experiment repeated on the contralateral side?

Answer: The reviewer is correct in pointing out that this issue wasn’t adequately addressed in the first version of the article. We have therefore decided to add an explanatory paragraph in the discussion section and to provide further raw data in tabular form as supplementary materials. To incorporate the raw data (without the depiction of any statistically significant results) into the current figure scheme, we believe, would make the current graphic presentations unwieldy and cluttered. Supplying the raw data as supplementary material is, we believe, a valid compromise between necessary translucency and a reader-friendly graphical illustration. .

Reaction: We added an explanatory paragraph in lines 589-593 and the supplementary tables.

  1. Can the effects of insulin on the kidney be reliably dissociated from the effects of hypoglycaemia?

Answer: This is a very legitimate question. Which we cannot answer in its entirety. By setting up and conducting the experiment, we tried to create a realistic representation of a possible insulin-induced hypoglycemia as well as a limited experiment. Due to the experimental setup, it is only possible to investigate effects on insulin-induced hypoglycemia under LFS of the kidney. Extended statements or a separation between the effects of hypoglycemia and an insulin effect cannot be shown in our experimental setup. We tried to represent this by a phase of normoglycemia under moderate insulin administration. Insulin was administered in both compared phases of our experiment.

Reaction: We have added a discussion of this issue in lines 450-455, but would also like the refer the reviewer to the already present acknowledgement and discussion of the potential influence of insulin in lines 438-450.

  1. Given that the standard errors are quite high, how comparable are results with results of former experiments?

Answer: We have already expanded on the differences in experimental models and other aspects (fasting, dosages, etc.) throughout the discussion and would like to refer the reviewer specifically to lines 427-443.

  1. Statistical methods could be better described, with p values & tests supplied for parameters that are stated to increase or decrease

Answer:

In the following, we’d like to describe our reasoning for the utilisation of the linear mixed effects model.

 In essence, a linear mixed-effects model is a statistical model that allows for both fixed and random effects.
Fixed Effects: These are the parameters of interest in a study that one expect to have a consistent effect. For example, this could be the type of treatment (drug vs. placebo).
Random Effects: These account for variability that is not of primary interest but might influence the outcome. This can include variability between subjects (inter-subject variability) or within subjects across time (intra-subject variability). For instance, if you are studying the effect of a drug on patients from different hospitals, the variability among hospitals could be a random effect.
The key advantages of linear mixed-effects models are:
Handling of Hierarchical or Clustered Data: In clinical studies, data often comes in nested forms, like multiple measurements from the same patient. Mixed-effects models can handle this clustering of data.
Flexibility in Dealing with Missing Data: They are robust in cases where some data points are missing, which is common in longitudinal studies.
Improved Estimation and Prediction: By incorporating random effects, these models provide more accurate estimates of the effects and better predictions for individual subjects.
In terms of implementation, one can define a model by specifying which variables are the fixed effects and which are random effects. The analysis will then give estimates of these effects, allowing to understand both the overall effect (fixed) and the variability (random) in the data.

Reaction: We added the relevant information to the figures throughout the manuscript where appropriate according to the reviewer’s recommendation. Furthermore – as already mentioned in response to another of the reviewer’s points – we have added further supplementary material in the sake of clarity and completeness of the raw statistical data. For easier accessibility and understanding, we have revised Figure 1.

  1. Comment re 150 min timeframe in the context of what is happening in the kidney - eg if steroid hormone alterations are relevant, this is a v short time.

Answer: You are right here too. The comment is absolutely correct. We designed our trial in such a way that it is also comparable in time with the preceding trials. For example, in a preliminary experiment we were able to demonstrate a time course of 150 min under insulin-induced hypoglycemia in prenatally HPAA-altered and non-HPAA-altered animals. There are findings on this, but not over a longer period of time. This is the background regarding the time span in the experiment.

  1. Comment on potential clinical application of findings, if relevant

Answer: Unfortunately, we cannot answer the legitimate question about clinical implications in humans. In the introduction we tried to explain why there are clinical references. We cannot give a firm assessment and a clinical reference. We addressed the issue of (clinical) transferability in more depth in our prior investigation of the effects of renal denervation on the renal capacity for gluconeogenesis during hypoglycemic states  - as there might be certain inherent risks for patient groups undergoing this procedure who are at simultaneous risks for hypoglycemia (from antidiabetic medications). This current study we intended to serve as a positive control study of our prior investigations and to have, firstly, a more elementary research focus. We have therefore decided against a further exploration of clinical transferability in the text.
